# Vascular Remodeling: The Multicellular Mechanisms of Pulmonary Hypertension

**DOI:** 10.3390/ijms26094265

**Published:** 2025-04-30

**Authors:** Jinjin Dai, Hongyang Chen, Jindong Fang, Shiguo Wu, Zhuangzhuang Jia

**Affiliations:** 1School of Basic Medical Sciences, Yunnan University of Chinese Medicine, Kunming 650500, China; daiijinn@163.com (J.D.); chenhyth@163.com (H.C.); m13649661060@163.com (J.F.); 2Yunnan Key Laboratory of Integrated Traditional Chinese and Western Medicine for Chronic Disease in Prevention and Treatment, Kunming 650500, China; 3Key Laboratory of Microcosmic Syndrome Differentiation, Education Department of Yunnan, Kunming 650500, China

**Keywords:** pulmonary hypertension, vascular remodeling, endothelial cells, smooth muscle cells, fibroblasts, cellular mechanisms

## Abstract

Pulmonary hypertension (PH) is a serious cardiovascular disease caused by a variety of pathogenic factors, which is characterized by increased pulmonary vascular resistance (PVR) and progressive elevation of mean pulmonary artery pressure (mPAP). This disease can lead to right ventricular hypertrophy and, in severe cases, right heart failure and even death. Vascular remodeling—a pathological modification involving aberrant vasoconstriction, cell proliferation, apoptosis resistance, and inflammation in the pulmonary vascular system—is a significant pathological hallmark of PH and a critical process in its progression. Recent studies have found that vascular remodeling involves the participation of a diversity of cellular pathological alterations, such as the dysfunction of pulmonary artery endothelial cells (PAECs), the proliferation and migration of pulmonary artery smooth muscle cells (PASMCs), the phenotypic differentiation of pulmonary artery fibroblasts, the inflammatory response of immune cells, and pericyte proliferation. This review focuses on the mechanisms and the intercellular crosstalk of these cells in the PH process, emphasizing recent advances in knowledge regarding cellular signaling pathways, inflammatory responses, apoptosis, and proliferation. To develop better treatments, a list of possible therapeutic approaches meant to slow down certain biological functions is provided, with the aim of providing new insights into the treatment of PH by simplifying the intricacies of these complex connections. In this review, comprehensive academic databases such as PubMed, Embase, Web of Science, and Google Scholar were systematically searched to discuss studies relevant to human and animal PH, with a focus on vascular remodeling in PH.

## 1. Introduction

Pulmonary hypertension (PH) is a chronic and progressive cardiovascular disease characterized by elevated blood pressure in the pulmonary arteries that can lead to significant morbidity and mortality; this can significantly impair the normal physiological function of patients’ pulmonary vascular system and their quality of life [1]. Referring to European data, PH was estimated at 5.8 cases per million adults annually, while its prevalence ranges from 47.6 to 54.7 cases per million adults. Notably, PH is disproportionately diagnosed among females and individuals in older adulthood [2]. In the latest European Society of Cardiology and European Respiratory Society (ESC/ERS) guidelines, it has been stated that the current hemodynamic definition of PH is characterized by a resting mean pulmonary arterial pressure exceeding 20 mm Hg, a pulmonary capillary wedge pressure of less than 15 mm Hg, and a pulmonary vascular resistance of at least 2 Wood units, as measured through right heart catheterization [3]. According to differences in clinical diagnosis, hemodynamics, and therapeutic regimens, pulmonary hypertension has been classified into five groups (Table 1): pulmonary arterial hypertension; pulmonary hypertension due to left-sided heart disease; pulmonary hypertension due to lung disease or hypoxia; chronic thromboembolic pulmonary hypertension; and pulmonary hypertension with unclear or multifactorial mechanisms [4].

The underlying cause of PH has yet to be definitively identified, but there is a widely accepted consensus that PH subtypes all share a fundamental pathological characteristic: vascular remodeling characterized by dysfunction of the inner membrane’s endothelial cells, proliferation and migration of middle membrane smooth muscle cells, and profibrotic responses of adventitial fibroblasts, as well as the involvement of multiple immune cells and progenitor/stem cells, ultimately leading to the thickening of the pulmonary artery vessel wall, narrowing of the lumen, and an increase in the resistance within the pulmonary vascular system which, in turn, leads to sustained constriction of healthy blood vessels, gradual stiffening of the vessel walls, and the formation of arterial vasculature plexiform lesions [5,6,7]. Remodeling processes in the lungs are triggered by factors including hypoxia, inflammation, shear stress, and genetic predisposition, leading to a cycle of deteriorating pulmonary hemodynamics and additional vascular changes.

By searching comprehensive academic databases such as PubMed, Embase, Web of Science, and Google Scholar, a comprehensive overview of the current understanding of pulmonary hypertension is provided, focusing on its definition, clinical manifestations, and epidemiological trends, as well as the underlying mechanisms of vascular remodeling. Through synthesizing recent research, this review endeavors to expound on the interaction between cellular crosstalk and intervention strategies designed to improve outcomes for patients with pulmonary hypertension, which is essential for both academics and physicians as it sets the stage for further studies and improvements in treatment.

## 2. Remodeling of the Vascular Wall

### 2.1. Endothelial Remodeling

Pulmonary hypertension is associated with impaired functioning of various cell types found within the pulmonary vasculature, specifically involving endothelial cells, which form a major part of the pulmonary artery endothelium [8]. The endothelium is the major determinant in maintaining its homeostasis. Functions of the normal endothelium include maintenance of vascular tone, regulation of leukocyte trafficking, luminal signaling, production of growth factors and cell signals with autocrine and paracrine properties, gas exchange, barrier function, and homeostasis; in addition, it has antithrombotic and anti-inflammatory characteristics [9]. Activation of endothelial cells by external stimuli (e.g., hypoxia, shear stress, and inflammation) leads to pulmonary vascular endothelial dysfunction, which is mainly characterized by aberrant proliferation, resistance to apoptosis, enhanced inflammatory signaling, and metabolic reprogramming [10] (Figure 1). Endothelial dysfunction is an important cause of vascular remodeling.

#### 2.1.1. Endothelial-Dysfunction-Related Factors

Among those external stimuli, hypoxia plays a dominant role. It has been found that among the many hypoxic factors, the common ones are mainly hypoxia-inducible factor-1 (HIF-1)- and hypoxia-inducible factor-2 (HIF-2)-mediated gene expression, but the role is still mainly dependent on HIF-1 [11]. Human epidermal growth factor receptor 3, (ErbB3) exhibited significant upregulation in the distal pulmonary arteries and pulmonary artery endothelial cells isolated from patients with PH. ErbB3 induced nuclear translocation of Y-box binding protein 1 (YB-1) and subsequently promoted HIF-1/2α transcription [12]. As a heterodimeric transcription factor consisting of HIF-α and HIF-β, HIF-1 is maintained at a normal level under aerobic conditions by proline hydroxylation, which contributes to its binding to the von Hippel–Lindau (VHL) ubiquitin ligase complex, where proteasomal degradation occurs [13]. In contrast, the rate of HIF hydroxylation is reduced under hypoxic conditions or VHL deficiency; HIF-α then accumulates and increases, dimerizes with HIF-β family members, and transcriptionally activates genes associated with angiogenesis as a result of the alteration of the normal physiological properties of blood vessels [11]. In addition, vascular endothelial growth factor (VEGF), a hypoxia-inducible factor, also known as vascular permeability factor, includes VEGF-A, -B, -C, -D, and placental growth factor and is not characteristically expressed in normal endothelial cells; notably, it is central to endothelial cell growth and survival factors [5]. According to a study, in PH, VEGF gene and protein expression is upregulated under hypoxic conditions. VEGF binds to the tyrosine kinase family of signaling receptors (e.g., flt-1, flk-1/KDR, and flt-4). In this context, flk-1 plays a major role in triggering a signaling cascade that leads to the tyrosine phosphorylation of phospholipase C gamma1 (PLCγ1), resulting in elevated intracellular calcium, activation of nitric oxide synthase (NOS), and production of nitric oxide (NO), which, in turn, affects vascular tone and stimulates angiogenesis by activating guanylate cyclase in endothelial cells to produce cyclic guanosine monophosphate (cGMP) [14,15,16]. This NO/cGMP cascade is thought to play a significant role in the vasoactive action of VEGF. Prostacyclin also affects pulmonary endothelial cell function by acting as a vasodilator, coupling to cell surface G protein receptors, promoting pulmonary artery dilation, and inhibiting vascular smooth muscle cell proliferation [16]. However, it was found that deletion of the raw material for prostacyclin, cyclooxygenase 1, leads to PH vascular remodeling and thrombus formation [17]. In contrast, endothelin-1 (ET-1)—a vasoconstrictor factor that often binds to G-protein-coupled receptors—was found to be elevated in the lungs of patients with PH, and a gene promoter polymorphism in ET-1 resulted in the loss of the binding site for the Kruppel-like factor (KLF), which further affected vascular remodeling [18].

#### 2.1.2. Hyperproliferation and Apoptosis of Endothelial Cells

The equilibrium between the growth and programmed death of endothelial cells is vital for preserving the integrity and operation of blood vessels. Endothelial cells (ECs) maintain organ homeostasis and assist in tissue regeneration. In adults, endothelial cells are largely quiescent under normal physiological conditions; interestingly, stimulation with some growth factors nevertheless promotes the generation of new endothelial cells [19]. VEGF is essential in the survival and proliferation of vascular endothelial cells. VEGF binds to two tyrosine kinase receptors—vascular endothelial growth factor receptor 1 (VEGFR-1; Flt-1) and VEGFR-2 (KDR/Flk-1)—on endothelial cells and undergoes a complex series of zymogen-activated reactions to promote the mitogenic and angiogenic functions of VEGF [20]. However, the presence of the Notch receptor—as an important activity regulator of VEGF—causes VEGF to be preferentially expressed with VEGFR2. SU5416 (commonly known as Sugen), a potent and selective VEGFR inhibitor, induces apoptosis of lung endothelial cells, lung small vessel injury, and gap enlargement [21]. Thus, ECs’ deletion of VEGFR2 or modulation with SU5416 leads to severe angio-obliterative PH, and VEGF overexpression attenuates the disease, suggesting a possible protective role for VEGF [22]. Moreover, it was found that SU5416 did not inhibit cell proliferation and, thus, VEGF receptor inhibition initially induced pulmonary artery endothelial cell (PAEC) apoptosis. Subsequent hypoxic exposure may stimulate the proliferation of a subgroup of anti-apoptotic PAECs [23]. Previous research has demonstrated that chronic blockade of VEGFR triggers the activation of forkhead box protein 1 (FOXO1), resulting in the expression of multiple tyrosine kinase receptors, including VEGFR2, ultimately leading to the emergence of anti-apoptotic ECs [24]. Excessive apoptosis can lead to endothelial dysfunction, impaired angiogenesis, and compromised tissue repair. Understanding this balance is vital for developing therapeutic strategies that enhance endothelial repair mechanisms while preventing excessive proliferation associated with disease states. Additionally, the forkhead box M1 (FOXM1) is an endothelial regeneration transcription factor that stimulates the proliferation of PAEC and pulmonary artery smooth muscle cells (PASMCs), promoting endothelial regeneration after lung tissue and vascular injury [25].

#### 2.1.3. Endothelial–Mesenchymal Transition

Endothelial–mesenchymal transition (EndMT) denotes the transformation of endothelial cells into mesenchymal cells [26]. Endothelial cells change from cobblestone-like to mesenchymal and fibroblastic phenotypes in response to inflammation, oxidative stress, and mechanical stress. They also acquire motility and contractile properties; in other words, endothelial cells gradually lose their distinct endothelial characteristics and take on a mesenchymal phenotype [27]. During EndMT, endothelial cells acquire invasiveness and migratory properties, detach from the cell layer, and invade the underlying tissue [28]. Throughout this transformation, endothelial cells lose endothelium-specific markers such as CD 31/platelet–endothelial cell adhesion molecule, vascular endothelial calreticulin, von Willebrand factor (vWF), and the expression of tyrosine kinase with immunoglobulin- and epidermal growth factor-like domain 1 (TIE1) and TIE2, which have immunoglobulin-like structural domains and epidermal growth factor (EGF)-like structural domains. In addition, they enhance mesenchymal cell markers such as N-adhesin, α-smooth muscle actin (α-SMA), waveform proteins, smooth muscle protein 22α (SM22α), fibroblast-specific protein-1 (FSP-1, also known as S100A4), and fibronectin expression [28]. Many studies have demonstrated that various signaling pathways and mediators, including vascular endothelial growth factor A (VEGFA), transforming growth factor β (TGFβ), inflammatory factors such as interleukin-1β (IL-1β), IL-6, and IL-10, and so on, bone morphogenetic proteins (BMPs), fibroblast growth factors (FGFs), epidermal growth factor receptor (EGFR), platelet-derived growth factor (PDGF), transcription factors (e.g., GATA4, Snail, Slug, and Twist1), and microRNA, all induce EndMT. Multiple additional signaling pathways, such as Wnt/β-linker protein signaling, NF-κB pathway, Notch signaling, and CaN/NFAT signaling, which have also been implicated in pulmonary hypertension, are also involved in EndMT [29,30]. It is well-known that EndMT is mainly responsible for exacerbating endothelial dysfunction and promoting pulmonary vascular remodeling, and, interestingly, it has been found that lineage tracing in PH can identify partial and complete EndMT states with different marker profiles. Partial-EndMT cells express endothelial progenitor cell markers such as endothelial progenitor protein-1 (PROM 1/CD 133) and CD 34 [31], which may be involved in neovascularization, angiogenesis, tissue repair, and regeneration [32,33,34], and are beneficial to some extent. Meanwhile, fully transformed cells express mesenchymal stem cell (MSC) markers such as stem cell antigen 1 (Sca 1) and endothelial glycoprotein (ENG/CD 105); as such, they may exacerbate vascular remodeling, fibrosis, and endothelial dysfunction [31]. However, the contribution of EndMT to disease progression is not fully understood but may alleviate PH symptoms and slow progression through the use of phosphodiesterase-5 inhibitors, prostacyclin analogs, and endothelin receptor antagonists [20]. Therefore, targeting the signaling pathway of EndMT is expected to provide new research directions in the treatment of PH.

#### 2.1.4. Endothelial Cell Activation and Thrombosis Formation

Abnormal activation of platelets and endothelial cells, together with the continued participation of coagulation factors, leads to the formation of in situ pulmonary artery thrombosis. This thrombotic lesion is a common pathologic feature in patients with PH. Endothelial cell activation is a precursor to thrombosis and is distinguished by the upregulation of adhesion molecules and proinflammatory cytokine secretion. This activation can be triggered by various factors, including oxidative stress, cytokines, and hypoxia [35,36,37]. Activated endothelial cells can contribute to blood clot formation by stimulating platelet activation via vWF, activating factor X, and initiating tissue factor (TF) activity [36]. Activation of platelets is mediated through the serine protease thrombin, adenosine diphosphate (ADP), collagen dimerization complex, and vWF [38], which leads to platelet aggregation. Platelet activation also increases the production of various adhesion molecules and receptors (e.g., selectin P and gp IIIa/IIb), as well as thromboxane A2 (TXA 2) [39]. The vasoconstrictor factor thromboxane A2 (elevated expression in patients with PH) has a proaggregatory effect, whereas the vasodilators NO and prostacyclin (reduced expression in patients with PH) inhibit aggregation [40]. Platelet-derived particles—which are circulating debris released from activated, damaged, or apoptotic platelets in the pulmonary vasculature of PH patients—can activate the coagulation cascade and stimulate thrombosis [41]. Recently, the omega-3 fatty acids eicosapentaenoic acid (EPA) and docosahexaenoic acid (DHA) have been found to reduce thrombogenicity by altering platelet function and decreasing the expression of inflammatory genes [42]. Interestingly, the chromatin and cytoplasmic enzymes formed after neutrophil death form a system of neutrophil extracellular traps, which activate platelets/endothelial cells and are thought to be mediators of thrombosis [36]. Therefore, targeting the signaling pathways mediated by platelet-activating factors is expected to alleviate intravascular thrombosis.

#### 2.1.5. Endothelial Cell Heterogeneity

Endothelial cell heterogeneity refers to cellular subpopulations of endothelial cells in different vascular beds with different phenotypic and functional characteristics. Relying on single-cell RNA sequencing (scRNA-seq) technology, data from scRNA-seq repositories of human lung tissues, and genetic genealogical tracing, it has been demonstrated that pulmonary vascular endothelial cells can be classified into multiple subtypes in the microvascular system, including PAECs, pulmonary venous endothelial cells, general capillary endothelial cells (gCap ECs), and air cell capillary endothelial cells (aCap ECs) [43,44,45]. Differences in these cells between individuals have been associated with the formation of the pulmonary artery vessel wall, but the mechanisms of action for these factors are not yet fully comprehended. It has recently been found that venous endothelial cells are involved in the formation of PAECs by migrating and transforming in the opposite direction of blood circulation, are the main source of neointimal expansion, and are the main endothelial subtype of arteriovenous malformation [46]. In addition to this, gCap ECs, which specialize in regulating the tone and function of vasodilation and act as stem/progenitor cells in the homeostasis and repair of capillaries, are also transformed and reprogrammed into arterial ECs during the development of PH, which may also be a new source of arterial endothelium [45]. Pneumocytes, as cells specific to lung tissue, contain airspaces and perform gas exchange independently of other endothelial subtypes, and they play a crucial role in leukocyte transport [43]. Along with these populations of endothelial cells, there is a population of endothelial tip cells that form vascular endothelial sprouts through the action of the CXC chemokine receptor 4a (CXCR-4a) and its ligand (C-X-C motif) ligand 12 (CXCL-12a) via the Notch/Wnt signaling pathways. These are, in turn, towed by stalk endothelial cells after the tip cells to promote vascular extension and remodeling [47,48]. Multiple endothelial cells are involved in the transformation of PAECs, which has an impact on vascular remodeling. However, it is unclear how PAECs maintain the stability of their cell populations during maturation and stabilization, and, thus, the in-depth study of endothelial cell heterogeneity is expected to reveal new entry points in the mechanism of vascular remodeling.

#### 2.1.6. Endothelial Cell Metabolism and Epigenetics

The effect of abnormal metabolism of ECs on pulmonary vascular remodeling has been demonstrated in the human pulmonary vascular system and PH models. Under aerobic conditions, vascular wall cells still exhibit the Warburg effect or abnormalities in glycolysis [49]. This metabolic shift is mainly regulated by the glycolytic enzyme 6-phosphofructo-2-kinase/fructose-2,6-biphosphatase 3 (PFKFB3), lactate dehydrogenase, and hexokinase, as well as the mitochondrial enzyme pyruvate dehydrogenase kinase [50]. In addition, PFKFB3 is a key regulator of glycolysis, and specific deletion of ECs inhibits PH development [51]. In addition, pyruvate kinase M2 (PKM2), a rate-limiting enzyme of glycolysis, participates in pyruvate formation to inhibit the glycolytic process and interacts with the protein JMJD8 to inhibit vessel formation [52]. *G6PD* (glucose-6-phosphate-dehydrogenase) is an X-linked gene that encodes the key enzyme in the glycolytic pathway. It has been found that the *N126D* mutant of *G6PD* (asparagine 126 mutated to aspartic acid), denoted as *G6pd^N126D^*, is associated with vascular remodeling in pulmonary hypertension. In *G6pd^N126D^ rats* administered with the vascular endothelial growth factor receptor blocker sugen-5416, the levels of plasminogen activator inhibitor-1, thrombin-antithrombin complex, and proinflammatory cytokine expression—including chemokine (C-C motif) ligand 3 (CCL3), CCL5, and CCL7—were elevated in the lungs, exacerbating inflammation, thrombosis, and hypertrophic pulmonary artery remodeling [53]. The pentose phosphate pathway is an important node of the glycolytic process linking other metabolic pathways. Mutations in the bone morphogenetic protein receptor 2 (BMPR2) gene are also involved in the metabolic process of PAECs. BMPR2 mutation may induce increased expression of the regulator of alternative splicing polypyrimidine tract-binding protein 1 (PTBP1) and the proglycolytic splicing product PKM2, which may promote the reprogramming of glycolysis [54]. Further studies have also revealed that the upregulation of β-catenin/Wnt and downregulation of peroxisome proliferator-activated receptor γ (PPAR-γ) through the TGF-β/Smad signaling pathway may accelerate glycolysis [55]. Therefore, the glycolytic pathway deserves to be further explored to find new therapeutic targets.

### 2.2. Media Remodeling

Smooth muscle cells (SMCs) are a crucial element of the vascular wall’s middle layer and play a crucial part in maintaining vascular homeostasis, as well as the typical vascular system. They are distributed throughout the vascular media layer and function to regulate blood pressure, vasoconstriction, arterial tone diameter, and blood flow distribution [56]. Under normal physiological conditions, SMCs exhibit a contractile phenotype marked by the presence of specific contractile proteins, such as smooth muscle actin and myosin. This contractile function is essential for the dynamic regulation of vascular resistance and is affected by diverse physiological stimuli, including shear stress, high pressure, and pulsatile blood flow [57]. In addition, SMCs are involved in the repair process after vascular injury, demonstrating their ability to adapt to different functional demands. The balance among the proliferation, migration, and apoptosis of SMCs is essential for the process of vascular homeostasis and remodeling [58]. However, in PH, vascular smooth muscle is more inclined to proliferate and migrate, and under normal conditions, there is only a coating of SMCs in non-myofibrillated distal pulmonary arterioles, whereas SMCs in more proximal arterioles are hypertrophied [5].

#### 2.2.1. Smooth Muscle Cell Phenotypic Transition

Under normal conditions, vascular smooth muscle cells (VSMCs) exhibit a static contractile phenotype that regulates the normal physiological function of the vascular system. In the early stages, VSMCs are migratory and proliferative and are relatively stable under normal conditions. When the vasculature is injured, contractile VSMCs are converted into a highly plastic synthetic phenotype. This process is considered to be the dedifferentiation of VSMCs (Figure 2), and the cellular phenotype changes from a spindle shape to an irregular shape, exhibiting high proliferation, migration, and accumulation of extracellular matrix components (e.g., collagen, elastin, and proteoglycans) [59]. VSMCs’ phenotypic conversion is an important step in the process of vascular regeneration and repair. Downregulation of contractile gene expression, cytoskeletal remodeling, and cellular reprogramming in synthetic VSMCs induced an increase in their proliferative capacity, and they combined with matrix metalloproteinases to promote extracellular matrix remodeling and the thickening of the vascular intima–media layer. Platelet-derived growth factor-BB (PDGF-BB), KLF4, miRNA, and CCL5 were found to regulate the dedifferentiation of VASM [60,61,62]. Nesfatin-1—an adipocytokine that elevates blood pressure—stimulates the conversion of contractile VSMCs into a synthetic phenotype and enhances the proliferation of VSMCs [63]. Although synthetic VSMCs are beneficial for the formation of neointima after vascular injury, their mechanism of endothelial remodeling is still worthy of in-depth study.

#### 2.2.2. Proliferation of Pulmonary Artery Smooth Muscle Cells

The proliferation of pulmonary artery smooth muscle cells is a prominent feature in PH. This process is impacted by various factors. Research has demonstrated that KLF4 functions as a regulatory factor in the process of vascular smooth muscle cell differentiation and proliferation during vascular remodeling development, and hypoxia factor (HIF1α)-induced PDGF-BB facilitates the high expression of KLF4 in SMCs. It has been found that anti-PDGF-BB inhibitors slowed down vascular remodeling by blocking the expression of PDGF [64]. Notch3, a member of the Notch family of receptors, induces VSMC proliferation and dedifferentiation. The Notch signaling pathway is essential for the identification of neointimal founder cells; a minority subset of SMCs marked by Notch3 could potentially act as the primary source of neointimal cells in the major population, and *Notch3-knockout mice* (Notch3−/−) are resistant to the development of hypoxia-induced pulmonary vascular remodeling and PH [65]. The fact that the transcription factor stuffing forkhead box protein is required for PASMC proliferation is well established. FOXO1 expression was downregulated in the pulmonary vasculature and PASMCs of human and experimental PH lungs, and the FOXM1 gene was upregulated in hypoxia-stimulated human pulmonary arterial smooth muscle cells (HPASMCs); thus, the specificity of the forkhead box protein may also be different in terms of the effects on vascular remodeling [66,67,68]. In contrast, BMP2 was shown to inhibit PASMC proliferation and promote apoptosis in PASMCs to protect against excessive vascular remodeling [69]. The balance of these factors is extremely important for the vascular wall, and exploring their intrinsic mechanisms of occurrence is important for our understanding of mesangial remodeling.

#### 2.2.3. Anti-Apoptosis in Smooth Muscle Cells and Reversal of Vascular Remodeling

The ability of SMCs to resist apoptosis is a key factor in vascular remodeling, especially under pathological conditions. The transient receptor potential canon (TRPC), especially TRPC3, involved in the anomalous store-operated Ca^2+^ entry mechanism results in PASMC phenotypes (exacerbated proliferation, enhanced migration, apoptosis resistance, and vasoconstriction); whereas, conversely, an increase in intracellular K^+^ ([K^+^] i) concentration improves apoptosis resistance and slows down intimal thickening [70]. Among the epigenetic factors, it has been found that the downregulation of *miR-204* activated the Src-STAT3-NFAT pathway of PASMC proliferation and apoptosis resistance in PH-PASMCs, which promoted neovascularization and aggravated the process of vascular remodeling [71], whereas miR-21-5p promoted apoptosis during hypoxia [72]. Further investigation is required to clarify the exact pathway through which transcription factor p53 triggers programmed cell death in SMCs via p21, inhibits SMC proliferation, and decreases neointimal development [73]. In addition, endothelial progenitor cell (EPC)-derived exosomes (Exos) (EPC-Exos) induced apoptosis in PASMCs and reversed the process of regeneration under hypoxic conditions [72]. These antiapoptotic mechanisms are essential for maintaining the number of SMCs during vascular injury and remodeling. However, excessive survival of SMCs can lead to pathological remodeling, such as neointimal hyperplasia and vascular occlusion. Strategies aimed at restoring the balance between SMC proliferation and apoptosis are being explored as potential therapeutic interventions for the reversal of vascular remodeling and re-establishment of regular vascular operation.

#### 2.2.4. Crosstalk Between Endothelial and Smooth Muscle Cells

The connection between ECs and VSMCs plays an important role in the development of the vascular system and the maintenance of vascular homeostasis. Especially in pathological conditions, ECs located in the intimal layer are directly exposed to the blood flow environment, and they are most vulnerable to stimulation by injury and inflammatory responses, releasing some vasoactive factors, such as NO, prostacyclin (PGI2), ET-1, and TXA2 [74], which mediate the influence on normal physiological functions and behaviors of VSMCs through paracrine effects. For example, NO produced by NO synthase in the endothelium diffuses to vascular smooth muscle cells, where it influences vascular tone and vascular smooth muscle cell proliferation by stimulating the formation of cGMP [75]. Extracellular vesicles act as carriers of intercellular information transfer, transporting proteins, lipids, and microRNAs to regulate the cellular phenotype [76]. It was found that PDGF-BB and TGF-β 1 expression was upregulated in endothelial–smooth muscle cells (ECs-SMCs) under low shear stress stimulation and played different roles in vascular remodeling, with PDGF-BB modulating the paracrine effect of ECs on SMCs and TGF-β1 acting as the feedback for the transition from SMCs to ECs. *MiR-663* is a potential biomarker for PH, targeting the TGF-β1/Smad2/3 pathway to decrease PDGF-BB-induced proliferation of PASMCs and, thereby, preventing pulmonary vascular remodeling and right ventricular hypertrophy in PH [77]. Moreover, in response to such adverse events, ECs and SMCs also produce soluble chemokines that recruit inflammatory cells, such as T cells, B cells, and macrophages, and promote the production of proliferative antiapoptotic factors, which affect vascular remodeling [78]. Understanding the intricate interactions between endothelial cells and SMCs is essential for developing therapeutic strategies aimed at preventing or reversing vascular remodeling.

### 2.3. Adventitial Remodeling

Fibroblasts are the most numerous cells in connective tissue. They are a major component of the adventitia of the vascular wall—the “damage-sensing component” of the vascular wall—and play a key role in maintaining tissue homeostasis and structural integrity. Fibroblasts are more inclined to synthesize and remodel the extracellular matrix (ECM) than endothelial cell dysfunction and smooth muscle cell proliferation and migration. At rest, fibroblasts are involved in the production of collagen, elastin, and glycoproteins, which enhance the tensile strength and flexibility of tissues. They are also involved in wound healing, migrating to the site of injury and promoting tissue repair by secreting growth factors and cytokines. The normal physiological function of fibroblasts is essential for tissue regeneration and repair, as well as for the maintenance of the ECM, which provides a scaffold for cell attachment and regulates cellular function through biochemical and mechanical signals [79,80,81]. Additionally, the transformation of myofibroblasts and the involvement of the immune response should not be overlooked (Figure 3).

#### 2.3.1. Activation of Fibroblasts

Activation of fibroblasts is a complex process that can be triggered by various stimuli, such as hypoxia, inflammatory cytokines, and growth factors. Increased contractile properties and ECM production are prominent hallmarks. More notably, hypoxia leads to the upregulation of α-SMA expression, which induces adventitial fibroblasts to differentiate into myofibroblasts, providing a fibrotic response for the vasculature to cope with the injury, and it is also thought to contribute to vascular remodeling. Interestingly, this increase in α-SMA level results from Galphai- and c-Jun NH_2_-terminal kinase (JNK)-dependent signaling pathways [82]. Premyelinated fibroblasts tend to proliferate more than SMCs in response to injury or stress, whereas postmyelinated fibroblasts exhibit SMC-like cellular qualities; however, their contractile properties are distinct from those of SMCs in that these vessels constrict gradually when exposed to vasoactive substances, and they do not fully relax in response to agents that cause vasodilation. At present, it is not clear whether fibroblasts can differentiate into SMCs [83,84,85]. Expression of fibroblast activation protein (FAP) is a distinctive feature of fibroblast activation, but a Type II membrane protein, CD26/DPP4, may act to inhibit fibroblast activation through the TGFβ signaling pathway in cases of chronic hypoxia [86]. Activated fibroblasts or myofibroblasts enhance the production of type-Ⅰ collagen and other ECM proteins (including fibronectin, tenascin-C, and elastin) in fibrotic tissues, which are essential for tissue repair and fibrosis, including the process of remodeling the vascular epithelium [87]. This increased ECM deposition is a double-edged sword: although it is critical for wound healing, excessive accumulation can lead to pathologic fibrosis and tissue stiffness, which can lead to a variety of fibrotic disorders, particularly with regard to vascular tone.

#### 2.3.2. Phenotypic Changes in Fibroblasts

It is notable that, in the context of injury, the phenotypic changes in fibroblasts—as the primary fibrogenic cell type—play an essential role in pathological tissue fibrosis. However, when their genetic profiles are analyzed using scRNA-seq datasets, a variety of fibroblast names is found, such as stromal cells, mesenchymal stem cells, myofibroblasts, and unknown mesenchymal cells [88]. Although there is ambiguity in the nomenclature of the cells, they can be distinguished from other cell classes by relying on extracellular markers of fibroblasts, such as vimentin, platelet-derived growth factor receptor A (PDGFR A), FAP, FSP1, and CD90 (Thy1) [89]. Fibroblasts exist in different tissues, during different developmental processes, and in different species, and they can be classified into different cell subpopulations. In lung tissues, the results of studying human lung fibroblast subpopulations revealed that the *CTHRC1^+^* pathological fibroblasts promote lung fibrosis in COVID-19 patients [90]. In response to injury stimuli, immune responses are involved. It has been found that in hypoxia-induced pulmonary vascular remodeling, the adventitia exhibits the earliest and most sustained immune response, and a particular proinflammatory phenotype of adventitial fibroblasts can stimulate macrophages via paracrine signals of IL6 and STAT3, HIF1, and CCAAT/enhancer binding protein β (C/EBPβ) pathways, as well as small extracellular vesicles (EVs), which participate in the modulation of the immune response and the activation of pro-inflammatory effects [91,92]. Fibroblasts with this pro-inflammatory phenotype secrete associated cytokines, including CXCL, CCL, CD40L, vascular cell adhesion molecule 1 (VCAM-1), interleukins, NO, prostaglandins, and TGFß, which activate the surrounding immune microenvironment [81]. Monocyte and macrophage migration and infiltration are facilitated by one of the primary chemokines known as monocyte chemoattractant protein-1 (MCP-1), which is also expressed in numerous cell types. Previous findings have shown that the production of MCP-1 by PH-/IPH-Fibs is tightly regulated by *miR-124*, and using histone deacetylase (HDAC) inhibitors to target the acetylation of *miR-124* is a promising strategy for reducing pro-inflammatory activation in fibroblasts [93].

#### 2.3.3. Crosstalk Among Fibroblasts, Endothelial Cells, and Smooth Muscle Cells

Dysfunction of the endothelium, migration and proliferation of the middle membrane, and deposition of matrix proteins in the adventitial fibroblasts lead to the onset of vascular remodeling lesions. Elevated vessel wall pressure and increased pulmonary artery (PA) flow trigger the mechanical stresses of vessel wall remodeling, leading to an increase in PAEC fluid shear stress, which accelerates the accumulation of pulmonary arterial adventitial fibroblasts (PAAFs) and the process of EndMT [94]. The role of communication between endothelial cells and adventitial fibroblasts is important in PH. ECs secrete ET-1 and IL-6, differentiating adventitial fibroblasts into myofibroblasts and, thereby, accelerating vascular remodeling. The increase in myofibroblasts results in elevated ECM stiffness, directly promoting the proliferation of PAECs [95]. The ECM gene of PAAFs is also expressed through mechanical stress activation, inducing excessive collagen accumulation, leading to an increase in ECM stiffness, and making vessels less compliant. Activated myofibroblasts also promote PASMC and PAEC proliferation, which, in turn, affects ECM stiffness and PA stiffness and increases vascular stiffness [96]. It has been found that in idiopathic pulmonary hypertension, secretions produced by PAAFs significantly alter phenotypic changes in normal smooth muscle cells. The expression of pentraxin 3 (PTX3), a prototypical member of the long pentameric protein family, occurs across a wide variety of cell types, including fibroblasts. Fibroblast growth factor (FGF), a member of the heparin-binding growth factor family, regulates SMC survival, proliferation, and migration. PTX3 binds to FGF2 and inhibits fibroblast growth factor receptor (FGFR) signaling, thereby impairing the pro-proliferative effects of fibroblast growth factor. Downregulation of PTX3 expression promotes PAAFs’ regulation of SMC proliferation; thus, the significance of PTX3 in vascular remodeling should not be overlooked [97]. ECs-SMC crosstalk is equally pivotal in preserving vascular homeostasis. Mediated by a hypoxia-induced mitogenic factor (HIMF), high-mobility group box 1 (HMGB1) and its receptor advanced glycation end products (RAGEs) participate in ECs-SMC communication. HMGB1, which is secreted by PAECs, is responsible for modulating autophagy and BMPR2 in PASMCs through paracrine mechanisms, promoting an overproliferative phenotype [98]. Understanding these interactions is prominent for expounding the mechanisms underlying tissue repair and the development of fibrosis.

### 2.4. Extracellular Matrix Remodeling

The extracellular matrix (ECM) is a highly organized molecular network composed of structural and nonstructural proteins, primarily including basement membrane (BM), collagen, elastin, laminin, fibronectin, tenascin-C, and proteoglycans [99]. These components provide essential stability and support for the structure of vascular wall cells and play a critical role in cellular and tissue homeostasis. In addition to supporting cells within the vascular layer, the ECM provides architectural and biochemical assistance to adjacent cells, serves as a landing site for inflammatory factors, and regulates intercellular signaling [99]. A normal ECM is vital for effective tissue repair, while abnormal regulation can lead to tissue remodeling. This remodeling occurs due to an imbalance between protein hydrolases (e.g., matrix metalloproteinases and elastase) and matrix metalloproteinase inhibitors, leading to increased collagen accumulation in perivascular and endovascular compartments, collagen cross-linking, and elastin breakdown. The equilibrium between extracellular matrix (ECM) synthesis and degradation is disrupted, ultimately compromising the structural integrity of the pulmonary vasculature [100,101].

#### 2.4.1. Stiffness of the Extracellular Matrix

The process of ECM remodeling is highly intricate and complex, and it is primarily driven by alterations in the equilibrium between collagen and elastin, which affects the deposition, degradation, and turnover of ECM proteins, ultimately resulting in the development of vascular fibrosis and vascular stiffness [102]. Matrix metalloproteinases (MMPs), as ECM turnover proteins, play a crucial role in the development of PH, coordinating with endogenous inhibitors to inhibit endogenous inhibitors and tissue inhibitors of metalloproteinase (TIMPs) and prevent excessive degradation of matrix proteins [100]. Recent findings that vascular stiffness not only precedes vascular remodeling but also appears in all forms of PH have led scholars to suggest that stiffness may serve as a predictor of disease progression and survival [103,104]. The normal pulmonary circulatory system is characterized by low pressure and high compliance. Collagen deposition thickens the vessel wall’s trilaminar membrane, decreasing proximal PA arterial compliance and increasing resistance to flow in the distal PA. Remodeling of the ECM leads to sclerosis of the proximal PA and distal vascular injury, and remodeling continues as pulse pressure and shear stress in the pulmonary vascular system are elevated, driving the development of PH from a positive feedback pathway [105]. However, research has increasingly demonstrated that ECM stiffness is not only a consequence but also a positive trigger of vascular remodeling. MicroRNAs (miRNAs) play a crucial role in mediating numerous cellular processes that involve both cell-to-cell and cell-to-matrix interactions. Activation of ECM hardening by the cotranscription factor Yap/TAZ induces the *microRNA-130/301* family to regulate the PPARγ-APOE-LRP8 axis to promote collagen deposition and lysyl oxidase (LOX)-dependent remodeling [106]. Matrix stiffening directly activates the proliferation of PASMCs and PAECs and drives the cyclooxygenase-2 (COX-2)/prostaglandin E2 (PGE2) pathway to promote fibroblast activation. Mechanical stress induced by vascular stiffness triggers a cascade of changes in ECs, PASMC proliferation, migration, and enhanced matrix deposition, which perpetuates vascular stiffness and creates self-sustaining feedback loops that exacerbate vascular remodeling in PH [107]. Therefore, weakening the stiffness of the ECM is beneficial for the inhibition of the development of the vascular remodeling process, but the study of the specific target mechanism remains to be refined.

#### 2.4.2. Inflammatory Response and the Extracellular Matrix

The inflammatory response has also been implicated as a cause of ECM remodeling in the pulmonary vasculature. Recruitment of various immune cells to the adventitia—such as macrophages, T cells, dendritic cells, and mast cells—affects vascular remodeling and collagen deposition through cell–cell contact or the delivery of inflammatory factors, especially in the intimal layer, where the highest levels of ECM collagen deposition are found [101]. In other studies, it was found that activated adventitial fibroblasts increased the expression of collagen, elastin, fibronectin, tenascin-C, and osteopontin in patients with PH. Conversely, tenascin-C and osteopontin increase the proliferation of fibroblasts and SMCs and assist in the transformation of myofibroblasts as a means of creating a stable closed loop that exacerbates vascular sclerosis [108]. The available results show that elastase expression is increased in PASMCs from patients with PH, causing degradation of the ECM and permitting stimulating or releasing the growth factors and signaling molecules from the ECM, such as FGF and TGF-β, to stimulate PASMC and fibroblast proliferation, giving rise to the formation of a stable loop of collagen, elastin, and other ECM structural proteins, such as those for increased deposition. In addition to this, the ECM provides a reservoir for these bioactive molecules to be stored in an inactive form [109]. More interestingly, inflammatory cytokines activate and recruit macrophages and neutrophils to secrete MMPs and elastase. The increase in MMPs is thought to be an adaptive response to the increased digestibility of the ECM at the endothelium and BM, and the protein hydrolysis products are proinflammatory and further activate the inflammatory response [110]. Involvement of the inflammatory response drives changes in the amount of ECM proteins, which drives a series of processes in vascular remodeling; thus, targeting ECM protein mechanisms may be beneficial in the treatment of vascular lesions.

### 2.5. Immune Cells

Research suggests that an impaired immune system plays a role in the formation of PH. Inflammation plays a significant role among various factors that contribute to the development of PH. Different inflammatory responses are prominent features of different types of PH, and although the exact mechanisms of immunity are unknown, the involvement of innate and adaptive immunity has been identified in patients with PH and in animal studies designed to simulate the condition of perivascular inflammation in lungs infiltrated by immune cells, as well as elevated levels of cytokines and chemokines, mainly due to T lymphocytes, B lymphocytes, macrophages, mast cells, dendritic cells (DCs), and neutrophils [111].

#### 2.5.1. T Cells

T cells play an important role in vascular remodeling as key players in the adaptive immune response, which include helper T cells (Th cells), regulatory T cells (Tregs), cytotoxic T lymphocytes (CTLs), and angiogenic T cells (Tang) [112]. Th cells generate pro-inflammatory responses, while TCRs mount a counterbalancing response to attain tolerance and thwart autoimmunity [113]. There are different subtypes of Th cells, including Th1, Th2, and Th17, and Th1 and Th17 cells can trigger an inflammatory response by producing interleukin (IL)-6, IL-2, IL-21, interferon-gamma, and tumor necrosis factor-alpha (TNF-α) [114]. Th17 cells secrete IL-17A, which promotes the migration of PASMCs and influences hypoxia-induced PH pathogenesis [115]. Conversely, Tregs inhibit inflammation in PH by secreting cytokines and chemokines such as IL-10, BMPR2, and CXCL12-CXCR4. In addition, in patients with CTD-PH, the Treg/Th17 ratio could be an important marker of PH prognosis [116]. Cytotoxic T cells have a potent capacity for cytolytic activity that triggers local inflammatory responses and are abnormally increased in peripheral lung tissues of patients with idiopathic and hereditary PH. Angiogenic T cells (Tang) expressing CD 31 and CXCR 4 have been shown to have a significant effect on angiogenesis [117].

#### 2.5.2. B Cells

B cells are the basis of humoral immunity, can differentiate into plasma cells, and are responsible for the production of autoantibodies. B cells do this through antigen presentation, production of various cytokines, and promotion of T effector cell differentiation [118]. B cells mainly secrete IL-6 and interferon-gamma (IFN-γ) pro-inflammatory factors, and IFN-γ-producing B cells have been found to have greater lung infiltrative properties in studies of the SSc-PH mouse model [119]. In addition to this, available studies have found that B cells have an effect on PH; B-cell-deficient *rats* have diminished susceptibility to severe monocrotaline (MCT)-induced or hypoxia-induced PH and pulmonary vascular remodeling [120]. Blockade of B cells by anti-CD 20 antibody or B-cell deficiency in *JH-KO rats* attenuates vascular remodeling in experimental PH [120]. Moreover, studies have demonstrated that B-cell removal therapy offers a balance of safety and effectiveness in the management of systemic sclerosis–PH and systemic lupus–PH [121].

#### 2.5.3. Macrophages

Macrophages, as a crucial component of the innate defense mechanism, play a crucial part in tissue homeostasis and immune system stability. Through powerful phagocytosis, macrophages encapsulate pathogens and foreign bodies and present them as antigens to T cells to facilitate differentiation and activation of the adaptive immune response [122]. It has been demonstrated that the development of the inflammation surrounding blood vessels in PH is strongly associated with macrophage recruitment. Excessive leukotriene B4 (LTB 4) synthesis by macrophages recruited around small pulmonary arteries induces apoptosis of PAECs, abnormal proliferation, and hypertrophy of PASMCs [123]. There are two main polarized phenotypes of macrophages: M1 and M2 macrophages. M1 macrophages are usually associated with pro-inflammatory responses and produce cytokines such as IL-6 and VEGF, and IL-6 has strong fibrillar properties. In contrast, M2 macrophages are involved in anti-inflammatory and tissue repair processes [124]. T follicular helper (T_FH_) cells that produce IL-21 were implicated in PH. The T_FH_/IL-21 axis has been found to enhance pulmonary vascular remodeling, which involves M2 macrophage polarization and recruitment [125]. Chemokine receptor 1 C-X3-C motif (CX3CR1)-knockout mice (CX3CR1^−^/^−^), however, exhibited an increase in macrophages and a change in macrophage polarization from M2 to M1. In addition, this study found that M2 macrophages promoted the growth and proliferation of PASMCs through the C-X3-C motif ligand 1/chemokine receptor 1 C-X3-C motif (CX3CL1/CX3CR1) axis and that CX3CR1 inhibition significantly reduced the proliferative effect of M2 macrophages on PASMCs, which was beneficial in slowing down the progression of PH [126]. Targeting and regulating the conversion mechanism of M1 and M2 macrophages is expected to provide new perspectives on therapies for PH.

#### 2.5.4. Mast Cells

Mast cells regulate inflammatory activity by releasing histamine and proteases, mainly during degranulation. Research has consistently demonstrated that the granule substances discharged by mast cells have a significant connection to the progression of PH. Chymotrypsin, a protease secreted by mast cells, indirectly relieves pulmonary vasoconstriction by activating the production of angiotensin II. (AngII.), endothelin-1 (ET-1), and MMPs, which inhibit PH and pulmonary vascular remodeling [127]. Trypsin-like enzymes, another protease released by mast cells, induce PASMC proliferation and migration and increase the synthesis of fibronectin and matrix metalloproteinase-2 in a protease-activated receptor 2/extracellular signal-regulated kinase 1 and 2 (PAR-2 /ERK1/2)-dependent manner to promote vascular remodeling [128]. In addition to this, mast cell activation produces lipid mediators such as lysophosphatidic acid (LPA), prostaglandin I2, LTB4, N-3 fatty acid epoxides, etc., which specifically regulate processes such as inflammation, angiogenesis, and fibrosis [129]. Mast cells have anti-inflammatory and immunosuppressive functions in addition to pro-inflammatory properties, but their specific mechanisms need to be demonstrated in further studies [130]. Since the number of mast cells and protein secretion are regulated by the c-kit tyrosine kinase receptor, the application of imatinib, a tyrosine kinase receptor inhibitor, as a mast cell stabilizer in the treatment of PH is still of interest [131].

#### 2.5.5. Dendritic Cells and Neutrophils

Dendritic cells (DCs) are specialized antigen-presenting cells that link the innate and adaptive immune systems, and they play a significant role in the immune response. DCs aggregate around remodeled pulmonary vessels and migrate to pulmonary lymph nodes and tertiary lymphoid organs (TLOs), resulting in the speculation that they may participate in the inflammatory response to PH with other immune cells [132]. Changes in DCs’ composition, maturation, and migration were found to be closely related to the architecture and capability of the pulmonary vascular system in *Tnfaip3^DNGR1-KO^ mice* [133]. DC homeostasis influences the development of PH, but the exact mechanism is not yet known.

Neutrophils are recognized to destroy pathogens through phagocytosis, degranulation, and the secretion of neutrophil extracellular traps (NETs) [134]. Fewer studies have been performed on the pathogenesis of neutrophils in PH, but it has been found that, in hypoxia- and MCT-induced *rats*, neutrophil levels are increased; furthermore, the expression levels of protein components of NETs, such as myeloperoxidase (MPO) and neutrophil elastase (NE), are upregulated in patients with PH [135]. NE exists in PASMCs and neointimal lesions in PH lungs, and released NE activates MMPs, promotes ECM degradation, and inhibits BMPR2 signaling, inducing SMC overproliferation and apoptosis in PAECs [136]. NETs also promote pro-inflammatory responses and angiogenesis in PAECs and exacerbate PH through the MPO/H_2_O_2_/NF-kB/TLR4 signaling pathway [137]. In a recent trial of elafin for PH, it was demonstrated that elafin improved PAEC homeostasis and reversed PH [136]. Emphasizing the role of NE versus NETs in neutrophils has some clinical potential.

### 2.6. Progenitor Cells and Stem Cells

Previous research has revealed the function of extravascular cells in PH and the in situ proliferation of resident vascular cells in the pathophysiology of PH. Through cellular ecological niches in the vessel wall, in the surrounding pulmonary mesenchyme, or distant tissues (primarily bone marrow), these undifferentiated vascular progenitor and stem cells—which possess the ability to self-renew and colonize—are mobilized and can transform new vascular cells in response to particular stimuli [138]. According to earlier research, BMPR2 mutations also control stem and progenitor cell differentiation. Endothelial progenitor cells (EPCs) are increased and have a hyperproliferative endothelial phenotype (CD45+/CD133+/c-kit+/CXCR4+) in PH patients with BMPR2 mutations, which impacts the formation of vascular networks and modifies the BMPR2 pathway [139]. Recent research has identified two distinct subpopulations of EPCs: early-growth EPCs originating from the hematopoietic lineage and late-growth EPCs generated from the endothelium lineage [140]. These EPCs have been discovered to have different modes of action in PH. The early-outgrowth EPCs release key proangiogenic factors such as CXCL12, VEGFA, and insulin-like growth factor-1, which primarily contribute to endothelial regeneration through paracrine actions [141]. On the other hand, the late-outgrowth EPCs are “more potent” progenitor cells than the early-outgrowth EPCs, as they are capable of producing mature endothelial progeny in vitro and participating in the development of new capillaries, although they occur in very small quantities in the blood. Due to their direct effects on blood vessel formation and the production of angiogenic factors, these cells reach their highest numbers following injury in order to repair damaged vessels [142]. Similar research has shown that the pulmonary vascular system of PH patients has markedly higher levels of some endothelial progenitor cell types that express c-kit, CD31, CD34, eNOS, caveolin-1, and vWF markers [143,144]. Thus, it is still interesting to see how progenitor cells and stem cells contribute to the development of PH pathogenesis.

### 2.7. Pericytes

Pericytes, located around endothelial cells, are multifunctional mural cells that have a role in regulating blood flow, are closely related to endothelial cells, and have been shown in PH models to be involved in early cell recruitment for vascular remodeling [145]. The PDGF-B/PDGFRβ signaling axis, which is also known as the endothelial–pericyte signaling axis, mediates this process [146]. Pericytes are SMC progenitors that have been shown in recent genetic lineages to induce pericyte differentiation into myofibroblasts through activation of the classical TGF-β signaling pathway [147]. On the other hand, pericytes migrate and proliferate through CXCL-12/CXCR7 signaling, acquiring functional and phenotypic aberrations under disease conditions [148]. More interestingly, an increased coverage of NG2^+^/3G5^+^ pericytes in the pulmonary vasculature was discovered in a PH model [148], and the progenitor cell subpopulation characterized by ATP Binding Cassette Subfamily G Member 2 (ABCG2) may also differentiate into α-SMA^+^ SMC/myofibroblasts and NG2^+^ pericytes [149]. Contradistinctively, decreased endogenous secretion of Wnt5a (a known Wnt/PCP ligand), a key mediator of pulmonary endothelial–pericyte interactions, inhibits pericyte recruitment to areas of neointimal growth, affecting vascular stabilization and regeneration of neovascularization [150]. The influence of pericytes on vascular remodeling cannot be ignored, and investigating the underlying pathological mechanisms of endothelial cell/pericyte interactions may provide new therapeutic ideas for PH.

## 3. Conclusions

PH is a cardiovascular disease that seriously affects human health, with pathological changes in the vascular system characterized by significant vascular remodeling. It is known that the proliferation and migration of ECs and SMCs, differentiation of fibroblasts, extracellular matrix deposition, the inflammatory response of immune cells, and pericyte proliferation are the pathological basis for the pathogenesis of PH; however, inter-cellular crosstalk, the functional heterogeneity of the cellular subsets, and the specific cell types involved are also crucial to the development of PH. The development of PH also harbors key pathogenic mechanisms that still need to be elucidated through further research. Although prostacyclin analogs, non-prostaglandin IP receptor agonists, HIF inhibitors, selective endothelin receptor antagonists, and phosphodiesterase 5 inhibitors have provided palliative care, none of the serviceable treatments for PH are curative and, at present, treatments for the side effects and the routes of administration in PH patient are still not well developed. Therefore, searching for new therapeutic targets is important. Clarifying other pathogenic mechanisms is clearly justified, and the development of new medicines designed to modulate and ameliorate the pathological remodeling process can be expected to broaden the therapeutic approaches to combat the severe morbidity and lethality associated with PH, ultimately leading to improved clinical treatment outcomes.

## Figures and Tables

**Figure 1 ijms-26-04265-f001:**
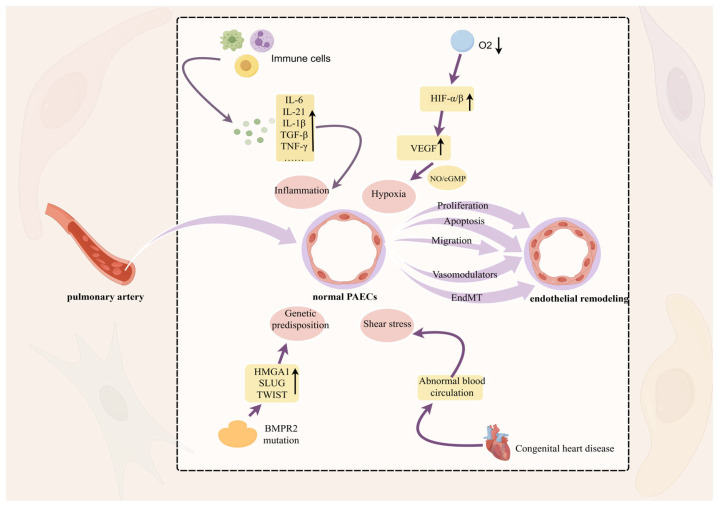
Remodeling of the pulmonary artery intima. Hypoxia, shear stress, epigenetic factors, and inflammation stimulate pathological changes in endothelial cell migration, proliferation, and apoptosis, leading to dysfunction and thickening of the vessel wall which, in turn, induces PH, with hypoxia playing a dominant role. Hypoxic conditions induce changes in the levels of HIF-α, which is involved in the regulation of proteasomal degradation and promotes angiogenesis through the NO/cGMP cascade of VEGF. In this figure, the upward arrow represents an increase in the substance and the downward arrow represents a decrease in the substance.

**Figure 2 ijms-26-04265-f002:**
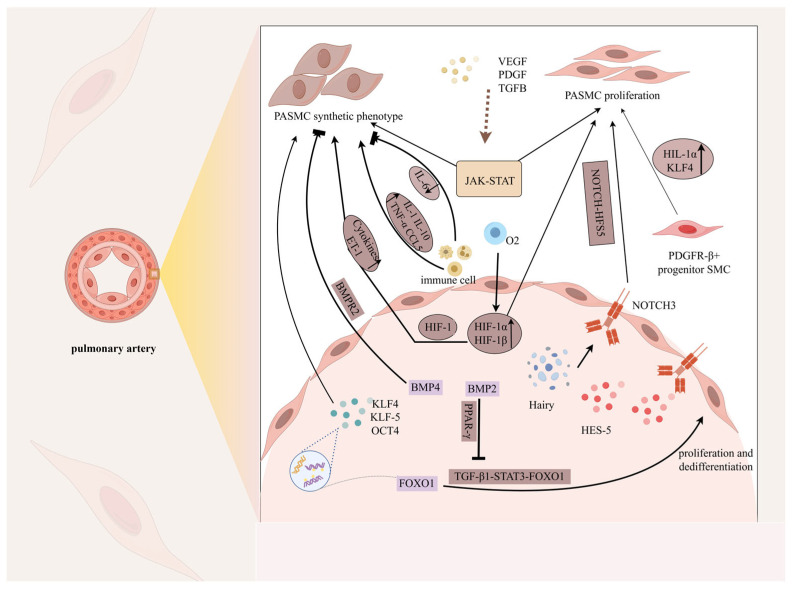
Phenotypic changes in pulmonary artery smooth muscle cells. Under the stimulation of mechanical injury, hypoxia, inflammatory factors (IL-1, IL-6, IL-10), various high-concentration growth factors (PDGF, TGF-β) and transcription factors (KLF, OCT, FOXO1), and signaling pathways such as JAK-STAT, NOTCH-HFS5, and TGF-β1-FOXO1, the phenotype of PASMCs gradually changed from a highly differentiated contractile phenotype to an undifferentiated rhombic synthetic cell phenotype, thereby enabling the cells to acquire the ability to proliferate, migrate, invade, synthesize, and secrete a large number of ECMs and special cytokines. Notably, proliferation is an important feature of their phenotypic change. In this figure, the pointed arrow represents promotion and the square arrow represents inhibition.

**Figure 3 ijms-26-04265-f003:**
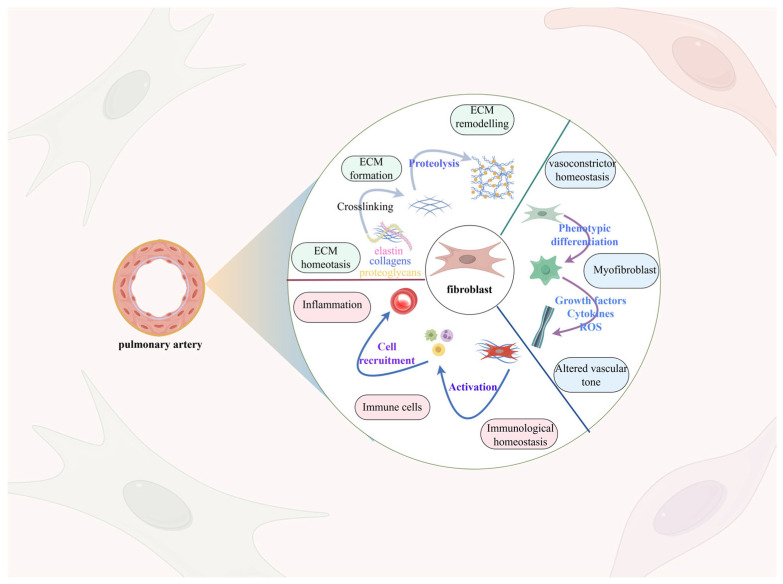
Remodeling of the pulmonary artery adventitia. Fibroblasts constitute a significant portion of the pulmonary artery adventitia, enabling vascular remodeling via processes including extracellular matrix deposition, fibroblast phenotypic alteration, and activation of the immune response. Normally, fibroblasts are responsible for producing collagen, elastin, and glycoproteins, but activated fibroblasts triggered by external factors such as hypoxia, immune factors, and shear stress boost the secretion of these proteins, resulting in the deposition of an extracellular matrix. Furthermore, the condition of hypoxia triggers an increase in the expression of α-smooth muscle actin (α-SMA), resulting in vascular fibroblasts transforming into a myofibroblast-like cell, thereby worsening the fibrosis of the vascular wall.

**Table 1 ijms-26-04265-t001:** An overview of the clinical classification and direction of medication for PH.

Group	Clinical Classification	Clinical Subtype	Therapeutic Drugs
1	Pulmonary arterial hypertension	Idiopathic, heritable, drug- and toxin-induced, venous/capillary (PVOD/PCH) involvement, viral (HIV) or parasitic disease (schistosomiasis), connective tissue disorders, liver cirrhosis, congenital heart disease, schistosomiasis, persistent PH of the newborn	CCBs (nifedipine, diltiazem, amlodipine), endothelin receptor antagonists, Ambrisentan, Bosentan, macitentan, PDE5i and guanylate cyclase stimulators (Sildenafifil, Tadalafifil, Riociguat), prostacyclin analogues and prostacyclin receptor agonists (Epoprostenol, Iloprost, Treprostinil, Beraprost, Selexipag)
2	PH associated with left heart failure	Left-sided atrial, ventricular, or valvular disease	Diuretics, PDE5i (e.g., Sildenafil)
3	PH associated with chronic hypoxemic lung disease	COPD, interstitial lung disease, obstructive sleep apnea, high altitude, developmental lung disorders	Treprostinil, PDE5is
4	PH associated with pulmonary artery obstructions	Pulmonary emboli, pulmonary hemangioma, pulmonary vasculitis, congenital pulmonary stenosis	VKAs, Riociguat, Treprostinil s.c., PDE5is (e.g., Sildenafifil), ERAs (e.g., Bosentan)
5	PH with unclear and/or multifactorial mechanisms	Sickle cell disease, sarcoidosis, metabolic disorders, renal failure, fibrosing mediastinitis	——

PVOD, pulmonary veno-occlusive disease; PCH, pulmonary capillary hemangiomatosis; CCB, calcium channel blocker; PDE5i, phosphodiesterase 5 inhibitor; COPD, chronic obstructive pulmonary disease; VKA, vitamin K antagonist; ERA, endothelin receptor antagonist.

## Data Availability

No new data were created or analyzed in this study. Data sharing is not applicable to this article.

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
