# Peer review of "Vascular Remodeling: The Multicellular Mechanisms of Pulmonary Hypertension"

_ijms, 2025, doi:10.3390/ijms26094265_

Round 1
Reviewer 1 Report
Comments and Suggestions for Authors
I would like to commend the authors for this thorough and well-structured review. The manuscript addresses a highly relevant topic and offers an in-depth and updated synthesis of the cellular and molecular mechanisms underlying pulmonary vascular remodeling in pulmonary hypertension (PH). The discussion is comprehensive, scientifically sound, and will be valuable to researchers and clinicians in the field.
The only suggestions I would like to make are the following:
-
Please consider including a brief Methods section, or at least a paragraph that describes the literature search strategy, the databases consulted, and the criteria for selecting studies. This addition would improve the transparency and rigor of the review process.
-
Language and style: While the manuscript is understandable overall, the English is at times technically dense or awkwardly constructed. A careful language revision by a native English speaker or a professional editing service is recommended to improve fluency and clarity.
These are minor issues that can be addressed with limited effort. I therefore recommend the manuscript for publication after minor revision.
Author Response
1. Please consider including a brief Methods section, or at least a paragraph that describes the literature search strategy, the databases consulted, and the criteria for selecting studies. This addition would improve the transparency and rigor of the review process.
Response 1: Thank you very much for your professional comments and suggestions on this review. Your comments on additional literature search strategies and screening criteria are important guidance for improving academic standardization. We fully agree with your views and will add the following contents to the sections of the abstract and introduction of the revised version upon request: L 29-32, "In this review, comprehensive academic databases such as PubMed, Embase, Web of Science, and Google Scholar were systematically searched to systematically discuss relevant studies on human and animal PH, with a focus on vascular remodeling in PH. " L 66-67, "By searching comprehensive academic databases such as PubMed, Embase, Web of Science, and Google Scholar, providing a comprehensive overview of the current understanding of pulmonary hypertension. " Thank you again for your professional comments.
2. Language and style: While the manuscript is understandable overall, the English is at times technically dense or awkwardly constructed. A careful language revision by a native English speaker or a professional editing service is recommended to improve fluency and clarity.
Response 2: We are very grateful for your professional comments on the language presentation of this review. We fully agree with your comments for professional language editing. We have accepted the English editing services provided by MDPI to systematically revise the language throughout the text, to standardize terminology and to ensure the scientific validity of the scholarly presentation, as we are not native speakers of English. Your valuable suggestions have greatly contributed to the refinement of the academic expression in this paper, and we thank you again for your rigorous monitoring of the linguistic standardization.

Reviewer 2 Report
Comments and Suggestions for Authors
The current article is comprehensive timely review on pathology of pulmonary hypertension with focus on vascular remodeling. The manuscript is well organized covering variety of aspect of vascular remodeling, containing some new insight. There are some minor comments for this review article.
Major points
- Table and Figures stand by themselves apart from the main text. Please cite these effectively in the text.
- There are some statements, sytax of which should be confirmed, as mentioned specifically in the section of Comments on the Quality of English Language.
- L 141-142: "SU5416 (commonly known as Sugen), ........ and gap enlargement" requires reference(s).
- The terms used for the genetically modified mice/rats is recommended to be made more specific and accurate. e.g., G6pdN126D (it is obvious for regulat scientists, but it is recommended to make some annotation for N126D), Notch3(-/-) mice (In this case, Notch3-knockout mice (Notch3-/-) is more accurate and specific)
- L 700-702: "Contradistinctively, the absence of Wnt5s ...... possibly increased the number of pericytes [149]" has two problem. One is the citation of Ref 149, which is not accurate. Ref 149 does not deal with anything related to Wnt5a. The study investigated the contribution of pericytes (vascular mural cells) and its SDF1 to vascular remodeling by using NG2-selective reporter mouse and NG2-positive cell-specific SDF1 knockout mouse. In relation to this, please re-confirm the accuracy of the citation of references throughout the manuscripy. The other is that the statement seems to be illogical. It state that absence of Wnt5a prevented pericyte recruitment and neovascularization, while such effect of absence of Wnt5a is described to be due to increased number of pericyte. Please clarify this confusion.
Minor points
- Abbreviations should be defined at their first appearance in the main text, even though they are defined in the Abstrac. Once defined, the abbreviations are recommended to use consistently. Please reconfirm the use of abbreviations throughout the manuscript. When defining the abbrevation, please use the consistent style. Namely, abbreivations in parethesis follows the spellout or spellout in parenthesis follows the abbreivations. Currently, both styles were employed. PAEC is to be defined on L 147. Please define PAAF, EC-AF, AFs, MFs, VR from L 460 though L 462. PAR-2 on 629 is to be derined. Definition and consistent use of abbreivation are recommended for "Dc", "DCs", "DC" on L 640-644. Please make "NETs" on L 649 to match the spellout. EPCs on L 667 is to be defined. Please also avoid use of abbreviations as much as possible for the non-expert readers especially in case of review article.
- L. 132: Please consider the necesity of use of EC for endothelial cells. If needed, please make consistent use throughout the manuscript.
- Line 75: "Endometrial" should read "Endothelial".
- Please confirm the accuracy of the term "tendonin/tendonin-C" and "bone-bridging proteins". No scientific terms for these were found in PubMed or NCBI database.
- L 458: Please confirm the accuracy of the term "from-like lesion."
- L 507: (TIMP) is recommended to put just after "endogenous inhibitors". And it should be "TIMPs."
- L 542: Please confirm the accuracy of "serine elastase", though it is scientifically correct, but unusual expression. It may be just "elastase" or "serine proteinase/protease."
There seem to be some syntax error or inaccurate logic in the following parts. Please reconfirm the accuracy of syntax and rewrite them if needed.
- L 364-365: "the eNOS/NO axis in ECs ....... proliferation and vasorelaxation."
- L 410-411: "Notably, in the present study, ........ a-SMA gene expression."
- L 475-476: " and we hypothesized that PTX3 plays a ........ a role in reversing the protective effect."
- L 493-500: "This remodeling results from an imbalance ......... ultimately compromising the structural integrity of the pulmonary vasculature." The sentence may be too long. In addition to the possible syntax error, this part also includes mis-use of parenthesis. A begining parentesis put before "e.g.," is not associated with the ending parenthesis. Please clarify this.
- L 548-554: "More interestingly, ........ further activate the inflammatory response [109]."
Author Response
Major points
1. Table and Figures stand by themselves apart from the main text. Please cite these effectively in the text.
Response 1: Thank you for pointing out the problems with the citation of figures and tables in the text, and we fully agree with you that a close correlation between figures and tables and the text is essential to ensure logical coherence of the research and smooth comprehension by the reader. We have made effective references to the figures and tables throughout the text, ensuring that they are numbered consecutively in the order in which they appear and that all references point to the actual figures and tables that exist. The revised contents are as follows: L 50: (Table 1), L 90: (Figure 1), L 306: (Figure 2), L 408: (Figure 3). Thank you again for your professional comments, which has greatly improved the academic standardization of this paper.
2. There are some statements, sytax of which should be confirmed, as mentioned specifically in the section of Comments on the Quality of English Language.
Response 2: Thank you very much for your valuable comments. In response to the section of Comments on the Quality of English Language, we have reviewed the initial literature and corrected the article-by-article content, and the revised content can be found in the section of Comments on the Quality of English Language. In addition, in order to make the expression of this review more rigorous and academic, we have accepted the English editing service provided by MDPI to systematically revise the whole review. Your professional suggestions are of great importance to the academic expression of this review, thank you again for your valuable comments.
3. L 141-142: "SU5416 (commonly known as Sugen), ........ and gap enlargement" requires reference(s).
Response 3: Thank you very much for your expert opinion on this review. We take this advice very seriously and have supplemented the specialized literature by searching databases such as PubMed and Web of Science, as detailed in reference [21]. Your suggestion has greatly enhanced the academic rigor of this paper, and we thank you again for your valuable comments.
4. The terms used for the genetically modified mice/rats is recommended to be made more specific and accurate. e.g., G6pdN126D (it is obvious for regulat scientists, but it is recommended to make some annotation for N126D), Notch3(-/-) mice (In this case, Notch3-knockout mice (Notch3-/-) is more accurate and specific)
Response 4: Thank you very much for your valuable comments on the standardization of terminology in this review. We have added detailed annotations for "G6pdN126D", and the revised formulation is as follows: L 266-269: "G6PD (glucose‐6‐phosphate‐dehydrogenase) is an X‐linked gene .......notated G6pd N126D", and for “Notch3(-/-) mice", we have unified it as L 339: "Notch3-knockout mice (Notch3-/-)". For other similar gene editing animal models, we have standardized the revision based on relevant literature. Thank you again for your professional comments.
5. L 700-702: "Contradistinctively, the absence of Wnt5s ...... possibly increased the number of pericytes [149]" has two problems. One is the citation of Ref 149, which is not accurate. Ref 149 does not deal with anything related to Wnt5a. The study investigated the contribution of pericytes (vascular mural cells) and its SDF1 to vascular remodeling by using NG2-selective reporter mouse and NG2-positive cell-specific SDF1 knockout mouse. In relation to this, please re-confirm the accuracy of the citation of references throughout the manuscripy. The other is that the statement seems to be illogical. It state that absence of Wnt5a prevented pericyte recruitment and neovascularization, while such effect of absence of Wnt5a is described to be due to increased number of pericyte. Please clarify this confusion.
Response 5: Thank you very much for your comprehensive and professional comments. Your feedback and suggestions were very helpful and insightful. We couldn't agree more with your question about the [149] literature citation and apologize for the error in citing the literature. We have re-searched the literature database to confirm that [150] correctly supports the discussion of the relationship between Wnt5a and pericytes. Due to the ambiguity in the meaning of the review owing to our poor presentation, we have adjusted the presentation as follows in the revised manuscript, L 715-718 : "Contradistinctively, decreased endogenous secretion of Wnt5a (a known Wnt/PCP ligand), a key mediator of pulmonary endothelial-pericyte interactions, inhibits pericyte recruitment to areas of neointimal growth, affecting vascular stabilization and regeneration of neovascularization[150]". For the original reference [149], we have deleted its incorrect citation in this review and ensured that it only retains the content of the relationship between Wnt5a and pericytes
Minor points
1. Abbreviations should be defined at their first appearance in the main text, even though they are defined in the Abstrac. Once defined, the abbreviations are recommended to use consistently. Please reconfirm the use of abbreviations throughout the manuscript. When defining the abbrevation, please use the consistent style. Namely, abbreivations in parethesis follows the spellout or spellout in parenthesis follows the abbreivations. Currently, both styles were employed. PAEC is to be defined on L 147. Please define PAAF, EC-AF, AFs, MFs, VR from L 460 though L 462. PAR-2 on 629 is to be derined. Definition and consistent use of abbreivation are recommended for "Dc", "DCs", "DC" on L 640-644. Please make "NETs" on L 649 to match the spellout. EPCs on L 667 is to be defined. Please also avoid use of abbreviations as much as possible for the non-expert readers especially in case of review article.
Response 1: Thank you very much for your valuable comments on the standardization of the use of acronyms in this review. We have referred to the normative requirements of acronyms and the full text of the acronyms combed, and standardize abbreviations throughout the text according to the format for the abbreviations in parenthesis follows the spelling. For the current mixed use of the labeling method, we have been revised one by one for the standard format to ensure that the definition of the consistency of the style. Finally, we have also supplemented and improved the section on abbreviations. The changes made to the above issues are mainly as follows: L 147 became L 152, PAEC is defined as pulmonary artery endothelial cell, L 460-462 became L 477-480, PAAF is defined as pulmonary arterial adventitial fibroblast, EC-AF has been revised to endothelial cells and adventitial fibroblasts, AFs is defined as adventitial fibroblasts, MFs has been replaced by myofibroblasts, VR is defined as vascular resistance, however, we realize that it is not appropriate to use it here and have changed it to vascular remodeling, L 629 became L 645, PAR-2 is defined as protease-activated receptor 2, L 640-644 became L 656-662, DCs is defined as dendritic cells, and we have standardized the use of DCs, L 649 became L 665, the spelling of NETs has been corrected to neutrophil extracellular traps, L 667 became L 684, EPCs has been defined as endothelial progenitor cells. Your patient and meticulous review of the academic rigor and readability of this paper has an important guiding significance, thank you again for your constructive comments.
2. L. 132: Please consider the necesity of use of EC for endothelial cells. If needed, please make consistent use throughout the manuscript.
Response 2: We are very grateful for your professional suggestions on consistency in terminology use. We fully recognize the importance of terminological standardization for the rigor of academic texts and have systematically checked and revised the use of endothelial cells abbreviations. We have standardized the use of ECs to represent endothelial cells and given definitions on first occurrence (L 137). Thank you again for your valuable comments.
3. Line 75: "Endometrial" should read "Endothelial".
Response 3: Thank you very much for pointing out the terminology error in the paper, we do apologize for the misuse of endometrial as endothelial due to our negligence (L 75 became L 79). We have systematically screened the entire text to ensure that all errors are corrected to the correct terminology, and we thank you again for your professional guidance.
4. Please confirm the accuracy of the term "tendonin/tendonin-C" and "bone-bridging proteins". No scientific terms for these were found in PubMed or NCBI database.
Response 4: Thank you for your critical review and valuable suggestions regarding the standardization of terminology in this review. We apologize for the irregular use of scientific terms due to our laxity. By searching the PubMed database, we have corrected the irregular use of "tendonin/tendonin-C" and "bone-bridging proteins". The term "tendonin/tendonin-C" has been corrected to "tenascin-C"(L 438, L 555-556), "bone-bridging proteins" has been corrected to "osteopontin"(L 555-556). Thank you again for pointing out these problems, and we will make serious revisions to ensure that the terminology in the paper is accurate and standardized.
5. L 458: Please confirm the accuracy of the term "from-like lesion."
Response 5: Thank you for your critical review of the accuracy of terminology in the paper. Regarding the use of the term "from-like lesion", we intended to describe the occurrence of lesions in which endothelial dysfunction and other factors lead to vascular remodeling, and as a non-native English speaker, we apologize for the irregularity in our use, and we have already corrected the "from-like lesion" to "vascular remodeling lesions"(L 458 became L 475). We would be serious about this and revise accordingly to ensure the accuracy and scientific quality of the terminology in the paper. accuracy and scientific validity to improve the quality of the paper. Thank you again for your professional comments.
6. L 507: (TIMP) is recommended to put just after "endogenous inhibitors". And it should be "TIMPs."
Response 6: We sincerely appreciated your valuable comments on the terminology presentation in this paper. Based on your suggestion, we have repositioned the labeling of "(TIMP)" after "endogenous inhibitor". For the format of "TIMP", we have searched the PubMed database accordingly and confirmed that the plural form “TIMPs” should be used when referring to the family of tissue inhibitors of metalloproteinases. We sincerely apologize for our writing error. In the revised manuscript, we have also fully labeled "Tissue Inhibitors of Metalloproteinases (TIMPs)" at the first occurrence to ensure accuracy and consistency in the use of the term (L 524-525). Thank you again for your professional guidance.
7. L 542: Please confirm the accuracy of "serine elastase", though it is scientifically correct, but unusual expression. It may be just "elastase" or "serine proteinase/protease."
Response 7: Thank you for your valuable input on standardizing the use of terminology. We have confirmed the correct writing of "serine elastase" by searching the relevant literature and standardized the "serine elastase" in the review to "elastase" (L 558、L 566). Thank you again for your professional comments on the rigor of the terminology used in this review.
Comments on the Quality of English Language
There seem to be some syntax error or inaccurate logic in the following parts. Please reconfirm the accuracy of syntax and rewrite them if needed.
- L 364-365: "the eNOS/NO axis in ECs ....... proliferation and vasorelaxation."
- L 410-411: "Notably, in the present study, ........ a-SMA gene expression."
- L 475-476: " and we hypothesized that PTX3 plays a ........ a role in reversing the protective effect."
- L 493-500: "This remodeling results from an imbalance ......... ultimately compromising the structural integrity of the pulmonary vasculature." The sentence may be too long. In addition to the possible syntax error, this part also includes mis-use of parenthesis. A begining parentesis put before "e.g.," is not associated with the ending parenthesis. Please clarify this.
- L 548-554: "More interestingly, ........ further activate the inflammatory response [109]."
Response: Thank you very much for your careful review and valuable suggestions on the linguistic quality of the paper. We fully agree that linguistic accuracy and logical coherence are essential for the academic presentation of this paper. We have accepted the language editing service provided by MDPI and have carried out a thorough check and systematic revision to address the issues you have pointed out. The revisions are as follows: 1. L 364-365 became L 378-380, "For example, Carbon monoxide (CO)/heme oxygenase (HO)-1 affects cGMP on SMC proliferation and vascular smooth muscle contraction by modulating endogenous NO and eNOS expression." 2. L 410-411 became L 426-429, "Notably, two signaling molecules, Gαi proteins and c-Jun NH2-terminal kinase (JNK), also increase the up-regulation of α-SMC expression and promote the process of fibroblast-to-myofibroblast transformation." 3. L 475-476 became L 495, " thus PTX3 may play a role in vascular remodeling by reversing remodeling." 4. L 493-500 became L 512-517, "This remodeling occurs due to an imbalance between protein hydrolases (e.g., matrix metalloproteinases, elastase) and matrix metalloproteinase inhibitors, leading to increased collagen accumulation in perivascular and endovascular compartments, collagen cross-linking, and elastin breakdown. The equilibrium between extracellular matrix (ECM) synthesis and degradation is disrupted, ultimately compromising the structural integrity of the pulmonary vasculature.", and we have standardized the parenthesis. 5. L 548-554 became L 564-568, "More interestingly, inflammatory cytokines activate and recruit macrophages and neutrophils to secrete MMPs and elastase. The increase in MMPs is thought to be an adaptive response to the increased digestibility of the ECM at the endothelium and BM, and the protein hydrolysis products are proinflammatory and further activate the inflammatory response. "

Round 2
Reviewer 2 Report
Comments and Suggestions for Authors
The manuscript has been improved somewhat. However, there are still critical issues.
- Newly added Reference 21 does not seem to be appropriate as a reference for the notion mentioned in the text (L.146-148). An appropriate reference should be cited.
- Newly added Reference 150 is the same article as that cited in the original manuscript. An appropriate reference should be cited.
- The following parts of the revised sentences where this reviewer had concern about syntax and logic still have linguistic problem. (1) L378-380, (2) L426-429, (3) L495
Author Response
1. Newly added Reference 21 does not seem to be appropriate as a reference for the notion mentioned in the text (L.146-148). An appropriate reference should be cited.
Response 1: Thank you very much for your valuable comments. We have carefully reviewed this section and strongly agree with you. Through a search of specialized databases, we have cited new reference [21] (L.146-148) and have removed the previous ones. Thank you again for your professional guidance.
2. Newly added Reference 150 is the same article as that cited in the original manuscript. An appropriate reference should be cited.
Response 2: Thank you very much for your patient and careful review. We apologize for our negligence in not revising the cited reference in a timely manner. After double-checking, we have ensured that the new reference [150] has been corrected. Thank you again for your valuable comments.
3. The following parts of the revised sentences where this reviewer had concern about syntax and logic still have linguistic problem. (1) L378-380, (2) L426-429, (3) L495
Response 3: Thank you very much for your professional comments. Because of our rather poor English expression, we have not made this section clear, for which we apologize. We have reviewed the logical connections in the context as well as the relevant references and have made several changes in the appropriate places. The changes are as follows: L378-380 became L382-385, "For example, NO produced by NO synthase in the endothelium diffuses to vascular smooth muscle cells, where it influences vascular tone and vascular smooth muscle cell proliferation by stimulating the formation of cGMP". We have also corrected the original references in order to make the content of this paragraph more clearly expressed [75]. L426-429 became L430-431, "interestingly, this increase in α-SMA level results from Galphai- and c-Jun NH2-terminal kinase (JNK)-dependent signaling pathways." L495 became l498, "thus, the significance of PTX3 in vascular remodeling should not be overlooked." In addition, we have accepted the English editing service provided by MDPI to systematically revise the whole review, and we have uploaded the certified document of the professional editing service in the Supplementary File(s) column as a supporting material for the improvement of the quality of English language. Thank you sincerely again for your constructive comments on this review.

Round 3
Reviewer 2 Report
Comments and Suggestions for Authors
The manuscript has been satisfactorily improved. There is no further comment.